# Comparison of Childhood Caries Levels between Children of Pediatric Dentists and Children of General Dentists: A Cross-Sectional Study

**DOI:** 10.3390/children10030452

**Published:** 2023-02-25

**Authors:** Sarit Naishlos, Sigalit Blumer, Sagit Nissan, Joseph Nissan, Johnny Kharouba

**Affiliations:** 1Department of Paediatric Dentistry, The Maurice and Gabriela Goldschleger School of Dental Medicine, The Faculty of Medicine, Tel Aviv University, Tel Aviv 6997801, Israel; 2Department of Oral-Rehabilitation, The Maurice and Gabriela Goldschleger School of Dental Medicine, The Faculty of Medicine, Tel Aviv University, Tel Aviv 6997801, Israel

**Keywords:** children dental caries, pediatric dentists, general dentists, fluoridated toothpaste, oral hygiene, nutrition habits

## Abstract

Caries development in children is a prevalent childhood disease. Factors affecting chronic teeth lesions include nutrition, parental involvement, and executing proper dental health attitude. Professional dentists are perceived as role models for the oral health and hygiene of their families. The purposes of the research were 1. To compare caries rates in the children of pediatric dentists and children of general dentists. 2. To compare children’s nutrition habits between pediatric dentists and general dentists. 3. To compare children’s oral health and hygiene between pediatric dentists and general dentists. 4. To compare children’s usage of fluoride-containing products between pediatric dentists and general dentists. A cross-sectional study was conducted by distributing self-reporting questionnaires to pediatric dentists and general dentists via the social media “snowball” platform. The following themes were surveyed: nutrition habits and oral hygiene of the children. The sample consisted of 176 participants. Children of pediatric dentists were found to have fewer cases of caries than children of general dentists (*p* = 0.018). Nutrition habits did not differ between the two groups. In addition, pediatric dentists reported that their children use more fluoridated toothpaste in comparison with general dentists. Professional training of pediatric dentists contributed to shaping the oral hygiene attitude of their children.

## 1. Introduction

Childhood caries are one of the most common and lingering diseases among children around the globe. Nearly all children have caries diagnosed at a certain stage ranging from initial to clinically manifested. For school-aged children in industrialized countries, caries incidence of 60–90% has been reported [1].

A Centers for Disease Control and Prevention (CDC) publication disclosed that among 5- to 10-year-old children, 20% have at least one decayed tooth due to negligence, while 13% of teenagers (aged 12 to 19 years old) are affected with ≥1 damaged, rotten tooth without dental care [2]. Moreover, it was stated that the likelihood of children aged 5 to 19 years from low-income families having cavities is significantly higher than their peers from higher-income households (25%, 11%, respectively) [2].

Childhood caries definition indicates a spectrum of lesions that span from the presence of a cavity in any primary tooth, to an affected tooth surface or a more deteriorated state of decay, to even missing an entire tooth. The clinical appearance of caries is uniquely characterized by the rapid development of the disease, which could hurt some teeth upon emergence in the oral cavity. Childhood caries is a multifactorial condition that involves the interaction of several elements that include cariogenic microorganisms, lingering exposure to fermentable carbohydrates due to inappropriate feeding habits, and a range of social variables that pertain both to the child and the parents’ behaviors [3].

Several etiological factors might be involved in the triggering and developing of dental caries. Etiologic factors can be clustered into three main groups.

Streptococcus mutans (SM) and Streptococcus sobrinus are the prevailing bacteria linked to dental caries in childhood [4]. SM breaks down carbohydrates into acidic end products that are involved in the process of tooth demineralization, which damages the tooth structure [5]. It has been reported that preschool children are at increased risk for developing caries upon the high presence of SM in their oral cavity [6].

Feeding practices of children implemented by their parents may determine an extended exposure of the teeth to the hazardous effect of fermentable carbohydrates, hence, resulting in Early Childhood Caries (ECC) [7].

Dental caries initially appears as dull white or brown spots on maxillary incisors along the gingival margin, which progress to the destruction of the crown, leading to root stumps [8].

Research shows that the development of childhood caries is attributed to biological and dietary elements, i.e., hosting microorganisms such as *Streptococci mutans* and repeated intake of high-sugar goods [9]. One of the most important factors that could influence the prevention or development of childhood caries is parental practices. Parents are well considered to be the main role model contributing to child development in the young childhood years in many domains, specifically in dental hygiene, which could serve as a buffer of childhood caries. Litt et al. [10] found that parents’ lower self-efficacy, dental knowledge, and parenting stress were all associated with increased rates of caries in preschool children. They also found that attributing problems to an external cause rather than accepting responsibility themselves (i.e., having an external, as opposed to an internal, locus of control) also affected childhood caries. Various parental risk factors were identified as related to childhood caries. First, the following demographic characteristics are associated with high childhood caries: lower social class and having a lower family income [11,12], single-parent families [13,14], being of higher birth order [15,16], low level of parents’ education [17,18,19], minority ethnicity [20,21], and living in a more deprived neighborhood [22].

In addition, studies found that specific children’s nutrition as executed by parents is correlated with high childhood caries, including frequent consumption of sweet foods and drinks, including snacks such as popcorn and dried fruits, and adding sugars to meals and fluids were components related to childhood cries [23,24]. Another factor associated with caries is youngsters’ habit of prolonged sipping of sugary drinks, such as juice or milk, through a bottle, due to lengthy access to fermentable carbohydrates [13]. One of the earliest factors contributing to childhood caries is the frequent intake of sweet drinks during the first six months of life, which may imply that parents’ unhealthy practices and poor dietary knowledge affect their children’s oral hygiene [14].

Another important aspect of parents’ involvement in their children’s oral hygiene is through daily supervision of tooth brushing and routine check-ups at dental clinics. A proactive attitude of parents toward instilling healthy dental behaviors would benefit their children’s oral hygiene; however, failing to do so could suggest that poor knowledge may lead to caries progress [25]. Moreover, an additional risk factor for childhood caries is a delayed initiation of tooth brushing, for example, after the age of one year old, in addition to a low frequency of brushing [26] and low time spent brushing [27]. However, lower caries rates were found to be related to good parental supervision of their children’s tooth brushing [13,26].

Another important risk factor for childhood caries is infrequent dental clinic visits. Children diagnosed with increased rates of caries were associated with missed annual appointments, the need for emergency dental treatment, and parents who infrequently visit the dentist’s office [27,28].

Finally, parental attitudes, knowledge, and beliefs regarding food and dental health were also found to be correlated with childhood caries. Parent beliefs represent their perception of reality, but this can be quite distinct from their actual knowledge and/or reality. Parents with the belief that “bad” teeth are inherited or parents with an external locus of control (i.e., a belief that they and their children are at the mercy of external events) had children with a higher risk of caries [29,30]. In addition, poor oral health knowledge and poor knowledge about fluoride were also associated with increased caries risk [13,22]. On the other hand, higher levels of knowledge were demonstrated in parents who requested sugar-free alternative medication [13] and who had lower monthly expenditure per household member on discretionary sugars [31] had children with a lower risk of caries.

Maternal health beliefs and practices related to oral health could affect the oral health habits that parents instruct their children. The Health Belief Model explains and predicts health behaviors by focusing on the attitudes and beliefs of individuals [32]. The Health Belief Model is a staged theory, with each step in the decision-making process being dependent on a previous belief. The theory predicts that an individual must believe that he or she is susceptible to a condition, that the condition is serious, that there is a successful intervention for the condition, and that he or she can cope with the barriers to adopting the intervention. Applying this theory to childhood dental caries, the parent must believe the child is susceptible to caries, that primary teeth are important, and that caries are a serious health threat. The parent also has to believe that dental caries can be prevented and must be willing to limit the child’s exposure to sugar snacks and practice oral hygiene for the child [33].

As presented in the previous section, parents have an enormous role in the development or prevention of childhood caries among their children. Specifically, the current work will focus on dentists as parents and the way that their profession could affect oral health among their children. Through their professional responsibility, dentists play a vital role in health promotion and preventive information dissemination. It is, therefore, essential that their oral health knowledge is good and that their oral health behavior reflects their understanding of the relevance of preventive dental procedures conforming to the expectation of the population. Specifically, dentists are expected to be good models for oral health behavior both for their patients and their children. They should also instruct their family members, friends, patients, and society to uphold good oral health [34,35].

To the best of our knowledge, no previous studies have been conducted to examine the effect of dentists on children’s oral health. However, previous studies examined dentists’ attitudes and behaviors regarding oral health. For example, researchers have found that the oral health attitudes and behaviors of dental students differed in the pre-clinical and clinical years. The clinical students, after their two years of pre-clinical training, are introduced to the preventive aspects of oral health only from the third year onwards; henceforth, their level of dental education can affect oral health behavior [34].

The main objective of the proposed study is to compare the children of pediatric dentists and children of general dentists in childhood caries levels and to understand potential explanations of possible differences in caries childhood levels between these groups. We aim to examine several objectives, including comparing children’s nutrition habits between pediatric dentists and general dentists, comparing children’s oral health and hygiene between pediatric dentists and general dentists, and comparing children’s usage of fluoride-containing products between pediatric dentists and general dentists.

## 2. Materials and Methods

### 2.1. Study Design

To compare the children of pediatric dentists and children of general dentists in childhood caries levels and in attitudes and behaviors, a cross-sectional study design was conducted. The study was conducted following the Declaration of Helsinki and approved by the Institutional Review Board of Tel-Aviv University (protocol code 000108-2, 28 March 2021). Self-report questionnaires were distributed to pediatric dentists and general dentists via different social media from April to July 2021.

### 2.2. Sample

The sample included two main groups—pediatric dentists and general dentists—that were further divided into different dental specialties who work both in private and public practice. All participants in this study are dentists who are parents. To assess sample size, we used G-power software under the following assumptions: type 1 error of 5% and minimum statistical power of 80%. In addition, we expect a moderate effect size of the difference in childhood caries levels between groups (Cohen’s d = 0.5), with a 1:1 ratio allocation to groups. Under these assumptions, the minimum sample size required for the study is 102 participants (51 for each group).

### 2.3. Measures and Variables

To gather data for this study, self-report questionnaires were administered, which included the following sections.

#### 2.3.1. Demographic Characteristics of Dentists

Gender, age, seniority, public/private practice, marital status, number of children, age of children.

#### 2.3.2. Childhood Caries Levels

Participants were asked to report the caries history of all their children, including frequency, severity, and number of decayed teeth.

#### 2.3.3. Nutrition Habits of Children

Participants were asked to report the nutrition habits of their children and describe the daily nutrition habits of their children in terms of types of food, quantities, and frequencies for each type of food. To measure these habits, we used The Children’s Eating Habits Questionnaire (CEHQ-FFQ), which consists of 43 food items on which parents or other caregivers were asked to report the number of meals the children usually consumed at home or other people’s homes, such as grandparents and friends, in a typical week of the previous month. Frequency categories ranged from 1 (Never) to 5 (Always). The 43 food items were clustered into 14 food groups: vegetables; fruits; drinks; breakfast cereals; milk; yogurt; fish; meat; eggs and mayonnaise; meat replacements; cheese; spreadable products; cereal products; snacks [36].

#### 2.3.4. Oral Hygiene of Children

The Oral Hygiene Habits Scale was filled in by the children [37]. The scale consisted of 10 items—6 items for Toothbrushing (e.g., How frequently do you brush your teeth?), and 4 items for Flossing (e.g., How much attention do you give to interdental cleaning?). Each item was rated on a Likert scale between 1 (the lowest level) and 5 (the highest level).

#### 2.3.5. Knowledge of Oral Health Scale

We used a 12-item scale to assess children’s oral health knowledge. The items included: (1) Eating sweets at bedtime can lead to cavities; (2) Frequently eating vegetables can lead to cavities; (3) Failure to brush teeth in the morning and evening can cause tooth decay; (4) Hard toothbrush is harmful to teeth; (5) Cavities should be treated in time; (6) Drinking milk is good for tooth development; (7) Cavities will affect the general health; (8) Toothbrushes should be replaced at least every three months; (9) The most important time to brush teeth is during nighttime; (10) Fluoride toothpaste can prevent cavities. Two possible responses, true (counted as 1 score) and false (counted as 0 scores), were used to answer the above statements, and the possible total scores ranged from 0 to 12. Higher scores indicated better knowledge regarding oral health [38].

### 2.4. Procedure

Participants (dentists) were recruited using the Medical Faculty at Tel Aviv University. Participants were asked to recruit other colleagues in a “snowball method”. After signing informed consent, participants were asked to respond to the study’s questionnaires. All dentists participated in this study voluntarily. Data was confidently kept, and no identifying details were collected.

### 2.5. Data Analysis

SPSS version 25 was used for data entry and analysis.

First, descriptive statistics were produced to study variables, using frequencies for categorical variables (e.g., gender) and means with standard deviations (SD) for continuous variables (e.g., age). Variables were tested for normal distributions using Shapiro–Wilk procedures to determine the use in parametric or non-parametric tests. Differences between pediatric dentists and general dentists were examined using chi-square tests for categorical variables and independent *t*-tests or Mann–Whitney tests for continuous variables (with normal or non-normal distributions, respectively). To assess risk factors for childhood caries, logistic regression was conducted, using the nutrition habits of children, the oral health of children, knowledge of the oral health scale, and demographic variables as predictors. Separate regressions were conducted for pediatric dentists and general dentists. The level of significance is 5%.

## 3. Results

### 3.1. Demographic and Professional Characteristics of the Sample

The sample consisted of 176 dentists, 31.3% pediatric dentists, and 68.6% general dentists. Table 1 presents the demographic and professional information of the sample and the differences between groups. As presented in Table 1, while most of the pediatric dentists were females (81.8%), only half of the general dentists were females (49.6%), *p* < 0.001. In addition, pediatric dentists were younger in comparison with general dentists (42.71 vs. 46.14, on average, *p* = 0.016) and with less seniority (15.46 vs. 18.96, on average, *p* = 0.019).

### 3.2. Caries in Children of Pediatric and General Dentists

Table 2 presents differences between groups in personal caries diagnoses. Results show that among pediatric dentists, there are fewer cases of children with caries in comparison with general dentists (0.65 vs. 1.13, on average, *p* = 0.018). In addition, fewer cases of significant caries were detected among pediatric dentists in comparison with general dentists (16.7% vs. 45.5%, *p* < 0.001). As for the dentists themselves, fewer cases of caries were detected among pediatric dentists in comparison with general dentists (63.6% vs. 76.9%, *p* = 0.068 as a marginal status). No differences were found between the groups in other variables.

### 3.3. Nutrition Habits Reported for Children of Pediatric and General Dentists

Table 3 presents differences between the children of pediatric and general dentists in nutrition habits. Results show that in most nutrition habits, no significant differences were found. However, it was found that pediatric dentists reported that their children consumed more fish in comparison with general dentists (2.89 vs. 2.58, on average, *p* = 0.03). A marginal difference was found regarding eggs, while pediatric dentists reported their children consumed more eggs in comparison with general dentists (3.81 vs. 3.59, on average, *p* = 0.13). No differences were found between the groups in other variables.

### 3.4. Dental Habits of Pediatric and General Dentists’ Children

Table 4 presents differences between the groups of paediatric and general dentists’ children in dental habits. In most dental habits, no significant differences were found between the two groups. However, a trend was found regarding using of fluoridated toothpaste. Specifically, paediatric dentists reported that their children use more fluoridated toothpaste in comparison with general dentists (4.73 vs. 4.47, on average, *p* = 0.17).

### 3.5. Pediatric and General Dentists’ Dental Knowledge

Paediatric dentists were found to gain more knowledge in comparison with general dentists (7.78 vs. 7.52, on average, *p* = 0.032, t = 1.86).

## 4. Discussion

As parents play a vital role in the development or prevention of childhood caries among their children, dentists hold the professional knowledge and skills to positively affect their children in preventing childhood caries. Therefore, paediatric dentists who specialize in the oral health behavior of children are hypothesized to demonstrate and promote childhood caries prevention among their patients, as well as their children.

This study aimed to compare the children of paediatric dentists and children of general dentists in childhood caries levels. Specifically, we hypothesized that paediatric dentists practice a healthier teeth lifestyle for their children because they have a higher awareness of the importance of teeth health for children.

In this cross-sectional study, oral health behavior was compared between the children of paediatric dentists with children of general dentists. Self-report questionnaires were distributed to paediatric dentists and general dentists via social media tools.

The present study results show that among paediatric dentists, fewer cases of children with caries were found in comparison with general dentists. Moreover, fewer cases of significant caries were detected among paediatric dentists in comparison with general dentists. This finding was also consistent among the dentists themselves, whilst fewer cases of caries were detected among paediatric dentists in comparison with general dentists. These findings implicate that professional training, alongside paediatric dentists’ focus on children’s caries, are the factors that could be referred to as the ones that positively influenced the lower rates of caries. Meaning the tools, knowledge, and skills that pediatric dentists provide to children in their clinics are similar to the ones they teach their children, probably more than their colleagues (general dentists). This set of oral health education tools is responsible for the children’s ability to avoid caries more effectively. Specifically, several of the main risk factors that have been documented for childhood caries, such as lower social class [11,12], low level of parents’ education [17,18,19], and living in a more deprived neighborhood [22], are relatively similar between the groups examined in this study (general vs. paediatric dentists). Hence, the results of this study rule out these alternative explanations for this finding.

In this study, nutrition habits were compared between pediatric and general dentists’ children. Overall, results showed that in most nutrition habits, both groups demonstrated similar nutrition habits. However, it was found that pediatric dentists reported that their children consume more fish and eggs in comparison with general dentists.

Finally, teeth lifestyle habits between the groups were examined. Overall, results showed similar habits. However, a trend was found regarding using of fluoridated toothpaste. Specifically, paediatric dentists reported that their children use more fluoridated toothpaste in comparison with general dentists. This finding is in line with previous studies that showed fluorides are the key element to successful caries prevention [39,40]. Fluorides are also effective as therapeutic agents in non-restorative caries treatment for the inactivation or arrest of caries lesions [41,42]. The use of fluoride toothpaste is one of the most important guidelines for parents, being convenient, inexpensive, culturally approved, and widespread [43].

The significant difference found in the current study could be explained by another important result achieved in this study. Specifically, paediatric dentists were found to be more knowledgeable regarding dental knowledge in comparison with general dentists. This difference is highly important since it emphasizes that the behavioral change paediatric dentists implement with their children is based on their knowledge regarding caries prevention among children, in addition to ways of conveying to children how to implement these oral health procedures.

Overall, the present study shows that the professional training of paediatric dentists contributes to their attitudes and shapes the oral health behavior of their children. However, the conclusions of this study are subjected to the following limitations. First, the data for this study have been gathered using a convenient sample. Therefore, the sample is not necessarily representative of the dental population in Israel. It is recommended that future studies will achieve a larger and more random study that will be representative of the dentist population. Second, this study relies on the professional training of dentists in Israel. However, oral health training might differ between countries. Hence, this study should be expanded and replicated in other countries. An additional limitation is that the caries levels were reported by the participating dentists and not assessed during a clinical examination, which would have reduced the risk of bias. Finally, it is a cross-sectional study and, therefore, is limited in its ability to conclude a causal relationship between pediatric dentists’ training and the oral health behavior of their children. Other alternative explanations might describe the lower rate of caries among children of pediatric dentists.

## Figures and Tables

**Table 1 children-10-00452-t001:** Demographic and professional characteristics of the sample.

Variable	Total Sample		Pediatric Dentists		General Dentists		X^2^	*t*	*p*
	N (%)	M (SD) ^1^	N (%)	M (SD)	N (%)	M (SD)			
Gender							16.32		<0.001
Male	71 (40.3)		10 (18.2)		61 (50.4)				
Female	105 (59.7)		45 (81.8)		60 (49.6)				
Age (years)		45.07 (9.28)		42.71 (8.09)		46.14 (9.62)	2.45		0.016
Marital status								0.152	0.696
Married	168 (95)		52 (94.5)		116 (95.9)				
Not married	8 (5)		3 (5.5)		5 (4.1)				
Number of children		2.62 (0.96)		2.73 (0.89)		2.61 (0.94)		0.76	0.444
Years of seniority		17.86 (9.57)		15.46 (8.61)		18.96 (9.82)		2.38	0.019
Field of Speciality									
Pedodontics	11 (18.6)		11 (91.7)		0			53.44	<0.001
Oral Medicine	8 (13.6)		0		8 (17.0)				
Oral Rehabilitation	8 (13.6)		0		8 (17.0)				
Orthodontics	10 (16.9)		1 (8.3)		9 (19.1)				
Maxillofacial Surgery	3 (5.1)		0		3 (6.4)				
Endodontics	8 (13.6)		0		8 (17.0)				
Periodontics	4 (6.8)		0		4 (8.5)				
Public Health	2 (3.4)		0		2 (4.3)				
Other	5 (8.5)		0		5 (10.6)				

^1^ Abbreviation: M: mean; SD: standard deviation.

**Table 2 children-10-00452-t002:** Comparison of caries information between pediatric and general dentists’ families.

	Pediatric Dentists	General Dentists	X^2^	*t*	*p*
	N (%)	N (%)			
Number of children with caries	0.65 (1.15)	1.13 (1.23)	2.44		0.018
Developmental dental defects	19 (34.5)	28 (23.1)		2.51	0.113
Significant caries of child	9 (16.7)	55 (45.5)		13.33	<0.001
Dental caries of the dentist	35 (63.6)	93 (76.9)		3.33	0.068
The severity of dental caries ^1^				3.745	0.154
Mild	25 (67.6)	48 (50.5)			
Moderate	8 (21.6)	37 (38.9)			
High	4 (10.8)	10 (10.5)			
Dental caries of dentist’s spouse	42 (76.4)	101 (83.5)		1.25	0.263
The severity of a spouse’s dental caries ^1^				0.677	0.713
Mild	19 (45.2)	51 (50.0)			
Moderate	17 (40.5)	41 (40.2)			
High	6 (14.3)	10 (9.8)			

^1^ Note: Mild:1-2 caries lesions; Moderate: caries level 3–4; High: >4 caries lesion.

**Table 3 children-10-00452-t003:** Comparison of nutrition habits between paediatric and general dentists’ children.

Food Item	Pediatric Dentists	General Dentists	*t*	*p*
	M (SD) ^1^	M (SD)		
Vegetables	3.87 (1.07)	3.92 (0.78)	0.36	0.71
Fruits	3.72 (1.02)	3.77 (0.83)	0.34	0.73
Sweet Beverages	2.10 (0.83)	2.11 (0.87)	0.04	0.96
Cereals	2.78 (1.03)	2.80 (1.01)	0.17	0.86
Dairy Products	3.72 (1.09)	3.54 (0.88)	1.17	0.24
Fish	2.89 (0.98)	2.58 (1.14)	1.84	0.03
Meat	3.60 (1.24)	3.57 (0.76)	0.14	0.88
Eggs	3.81 (0.98)	3.59 (0.86)	1.52	0.13
Salty Snacks	2.50 (0.92)	2.61 (1.01)	0.69	0.49
Sweets	2.78 (0.91)	2.80 (0.94)	0.13	0.89

^1^ Abbreviations: M: mean; SD: standard deviation.

**Table 4 children-10-00452-t004:** Comparison of dental habits between paediatric and general dentists’ children.

Dental Habits	Pediatric Dentists	General Dentists	*t*	*p*
	M (SD) ^1^	M (SD)		
Frequency of Check-Ups/Preventive treatment	2.94 (0.93)	3.00 (1.13)	0.37	0.71
Frequency of not brushing teeth	1.61 (0.76)	1.60 (0.84)	0.11	0.91
Use of fluoridated toothpaste	4.73 (0.97)	4.47 (1.17)	1.37	0.17
Attention while toothbrushing	3.78 (0.91)	3.82 (0.96)	0.26	0.79
Time in each tooth brushing (*)	2.46 (1.01)	2.41 (0.98)	0.28	0.77
Frequency of changing the toothbrush	3.40 (1.06)	3.36 (1.18)	0.19	0.85

^1^ Abbreviation: M: mean; SD: standard deviation. * The values represent the sum of the answers in the Likert scale.

## Data Availability

The data presented in this study are available on request from the corresponding author.

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
