# Peer review of "Comparison of Childhood Caries Levels between Children of Pediatric Dentists and Children of General Dentists: A Cross-Sectional Study"

_children, 2023, doi:10.3390/children10030452_

Round 1
Reviewer 1 Report
Methodological Biases exist
(The Authors must see my remarks)

Author Response
We thank you for taking the time to review our manuscript and for the comments.
Below is a point-by-point response to the points raised.
Reviewer 1
Header: Type of article is Research
Title: A cross-sectional study was added
Introduction:
subsections were omitted as per the reviewer’s suggestion
Reference 2 was added to line 44.
Numbers were omitted from the study objectives lines (145-151).
study objectives
2.2 Sample inclusion/exclusion criteria:
" All participants in this study were dentists who are parents" line 163
2.3.1 Measures and Variables
Sex was replaced by Gender as per reviewer’s request.
2.5 Data analysis sex was replaced by “gender."
Table 1 Sex was replaced by Gender.
Discussion:
risk of bias was added to lines 329-331
References:
Reference 36-page number was corrected.
- Huybrechts, I.; Börnhorst, C.; Pala, V.; Moreno, L.A.; Barba, G.; Lissner, L.; Fraterman, A.; Veidebaum, T.; Hebestreit, A.; Sieri, S.; Ottevaere, C. Tornaritis, M.; Molnar, D.; Ahrens, W.; De Henauw, S.; IDEFICS Consortium. Evaluation of the Children's Eating Habits Questionnaire used in the IDEFICS study by relating urinary calcium and potassium to milk consumption frequencies among European children. Int J Obes (Lond) 2011, 35(1), S69-78. doi: 10.1038/ijo.2011.37
Rephrasing
Line 20 "… the oral health and hygiene of their families"
Line 21 "… purposes of the research were"
Line 27 "… nutrition habits and oral hygiene of the children"
Line 31-32 "…oral hygiene attitude"
Line 38 "… common and lingering"
Line 39 " …around the world"
Line 30-41 "Nearly all children are would have caries diagnosed at certain stage ranging from initial to clinically manifested. For school aged children in industrialized countries caries incidence of 60%-90% has been reported"
Lines 42-47 "The Centers for Disease Control and Prevention (CDC) publication, disclosed that among 5 to 10 years old children, 20% have at least one decayed tooth due to negligence, while 13% of teenagers (aged of 12 to 19 years old) are affected with ≥ 1 damaged rotten tooth without dental care [2]. Moreover, it was stated that the likelihood of children aged 5 to 19 years from low-income families to have cavities is significantly higher than their peers from higher-income households (25% vs. 11% respectively) [2]."
Line 48-55 " Childhood caries definition indicates a spectrum of lesions that spans from the presence of a cavity in any primary tooth, to an affected tooth surface or a more deteriorated state of decay, to even missing an entire tooth. The clinical appearance of caries is uniquely characterized by the rapid development of the disease, which could hurt some teeth upon emergence in the oral cavity. Childhood caries is a multifactorial condition that involves the interaction of several elements that include cariogenic microorganisms, lingering exposure to fermentable carbohydrates due to inappropriate feeding habits, and a range of social variables that pertain both to the child and parents' behaviors [3]."
Line 58-59 " …the prevailing bacteria linked to"
Line 59-60 " SM breaks down carbohydrates to acidic end products that are involved the process of tooth demineralization which damages the tooth structure "
Line 61-62 " It has been reported that preschool children are in increased risk for developing caries upon high presence of SM in their oral cavity "
Line 63-64 "may determine an extended exposure of the teeth to the hazardous effect of fermentable carbohydrates hence resulting in"
Line 69-71 "Research shows that the development of childhood caries is attributed to biological and dietary elements i.e., hosting microorganisms such as Streptococci mutans and repeated intake of high sugar goods"
Line 73 " the main role model contributing to child development in the young"
Line 87-88 " including snacks such as popcorn and dried fruits, and adding sugars to meals and fluids were components related to childhood cries"
Line 88-90 "Another factor associated with caries is youngsters' habit of prolonged sipping of sugary drinks such juice or milk through a bottle, due to lengthy access to fermentable carbohydrates.
Line 90-92 " One of the earliest factors contributing to childhood caries is frequent intake of sweet drinks during the first six months of life, which may imply parents' unhealthy practices and poor dietary knowledge affect their children oral hygiene"
Line 94-95 "Another important aspect of parents' involvement in their children oral hygiene is through daily supervision of tooth brushing and routine checkups at dental clinics"
Line 95-98 "Proactive attitude of parents toward installing healthy dental behaviors would benefit their children oral hygiene however failing to do so could suggest that poor knowledge may lead to caries progress
Line 98-99 "Moreover, an additional risk factor for childhood caries is a delayed initiation of tooth brushing for example after the age of one year old.
Line 100-101 "lower caries rates were found to be related to good parental supervision of their children tooth brushing.
Line 103-106… infrequent dental clinic visits. Children diagnosed with the condition of increased rates of caries were associated with missed annual appointments, need of emergency dental treatment and parents who infrequently visit the dentist office"
Reviewer 2 Report
Comments to the author
Thank you for inviting me to review the manuscript entitled “Comparison of childhool caries levels between children of pediatric dentists and children of general dentists”.
The study presents a comparison between children of pediatric dentists and children of general dentists in terms of rates of caries, nutrition habits and oral health of the children.
The manuscript is very well-written, and the results are interesting. I congratulate with the authors for such an interesting idea and a very well-conducted study.
I support the publication with no corrections.
Author Response
Thank you for your tour fruitful comments.
Reviewer 2
Does the introduction provide sufficient background and include all relevant references?
Can be improved.
Improved
Reviewer 3 Report
Dear authors,
Congratulations to your study! It was interesting to read your manuscript on the results of your self-reporting questionnaire provided to pediatric dentists and general dental practitioners reporting about their children’s caries prevalence, nutrition habits, oral hygiene, and oral health knowledge. I recommend considering the following suggestions during revision:
1. Abstract:
- When reporting about the results, could you please add the numbers and p-values to the abstract?
2. Introduction:
- Please clarify what is meant by childhood caries. Do you refer to early childhood caries (ECC) since many of your references are about ECC; or are carious permanent teeth also included in this term because the first paragraph also includes prevalence data for older children and adolescents?
- I missed mentioning the role of fluorides in caries prevention. Perhaps you could add this to your introduction.
3. Methods:
- Paragraph 2.3.4: Could you imagine renaming the heading because the questions mentioned were related to oral hygiene rather than to oral health.
4. Results:
- How was the response rate for your questionnaire?
- Table 1: Have you collected any information about the children’s age?
- Table 2: What does significant caries mean? How is mild / moderate / high for the severity of dental caries and the severity of spouse’s dental caries defined? Please add a legend for the abbreviations used as you have done it for the other tables.
- Paragraph 3.5: Are these the results of the children’s oral health knowledge? From the methods section I thought the children had answered these questions but from the results it appears as if the parents (the dentists) had responded to these items. Could you please clarify this?
5. Discussion:
- When addressing the limitations, it should be added that the caries levels were reported by the participating dentists and not assessed during a clinical examination, which would have reduced a risk of bias.
Kind regards!
Author Response
Reviewer 3
We thank the reviewer for taking the time to review our manuscript and for the comments. Below is a point-by-point response to the points raised.
1 abstract
When reporting the results, could you please add the numbers and p-values to the abstract?
P=0.018 (Table 2) was added to the abstract. (Line 28)
2 introduction:
- Please clarify what is meant by childhood caries. Do you refer to early childhood caries (ECC) since many of your references are about ECC; or are carious permanent teeth also included in this term because the first paragraph also includes prevalence data for older children and adolescents
In the first paragraph we meant caries in children including permanent teeth in children above 6 years of old (ref. 1,2)
The second paragraph we wrote about ECC (Corrected, line 46,49)
- I missed mentioning the role of fluorides in caries prevention. Perhaps you could add this to your introduction.
Mentioned lines 114-115, Ref.13,22.
3 Methods:
Paragraph 2.3.4: Could you imagine renaming the heading because the questions mentioned were related to oral hygiene rather than to oral health
Renamed to Oral hygiene of children (line 200)
4 Results:
- How was the response rate for your questionnaire?
Unfortunately, we did not consider this important point.
- Table 1: Have you collected any information about the children’s age?
No. we missed these important details
- Table 2: What does significant caries mean? How is mild / moderate / high for the severity of dental caries and the severity of spouse’s dental caries defined? Please add a legend for the abbreviations used as you have done it for the other tables
Added legend to Table 2
"Mild caries level:1-2 caries lesions
Moderate caries level 3-4
High caries level >4 caries lesions."
- Paragraph 3.5: Are these the results of the children’s oral health knowledge? From the methods section I thought the children had answered these questions but from the results it appears as if the parents (the dentists) had responded to these items. Could you please clarify this?
In lines 168-169 (methods), it is mentioned:
"Self-report questionnaires were distributed to pediatric dentists and general dentists via different social media from April to July 2021."
5 Discussion:
- When addressing the limitations, it should be added that the caries levels were reported by the participating dentists and not assessed during a clinical examination, which would have reduced a risk of bias.
That's true. we added this comment to the limitations. (Lines 339-341)